# Control System Design of an Underactuated Dynamic Body Weight Support System Using Its Stability

**DOI:** 10.3390/s21155051

**Published:** 2021-07-26

**Authors:** Grzegorz Gembalczyk, Piotr Gierlak, Slawomir Duda

**Affiliations:** 1Department of Theoretical and Applied Mechanics, Faculty of Mechanical Engineering, Silesian University of Technology, Akademicka 2A, 44-100 Gliwice, Poland; slawomir.duda@polsl.pl; 2Department of Applied Mechanics and Robotics, Faculty of Mechanical Engineering and Aeronautics, Rzeszow University of Technology, 35-959 Rzeszów, Poland; pgierlak@prz.edu.pl

**Keywords:** stability, adaptive control, underactuated mechanical system, control theory, gait rehabilitation system, model-based control

## Abstract

This paper discusses the stability of systems controlling patient body weight support systems which are used in gait re-education. These devices belong to the class of underactuated mechanical systems. This is due to the application of elastic shock-absorbing connections between the active part of the system and the passive part which impacts the patient. The model takes into account properties of the system, such as inertia, attenuation and susceptibility to the elements. Stability is an essential property of the system due to human–device interaction. In order to demonstrate stability, Lyapunov’s theory of stability, which is based on the model of system dynamics, was applied. The stability of the control system based on a model that requires knowledge of the structure and parameters of the equations of motion was demonstrated. Due to inaccuracies in the modeling of the rope (one of the basic elements of the device), an adaptive control system was introduced and its stability was also proved. The authors conducted simulation and experimental tests that illustrate the functionality of the analyzed control systems.

## 1. Introduction

For a patient who is post-stroke, post-trauma or post-surgery, the effectiveness of their therapeutic treatment is increased by starting rehabilitation as early as possible. The effectiveness of rehabilitation depends, to a great extent, on the intensity of the exercises performed. As a result, modern rehabilitation devices that operate with a high level of precision and reliability are becoming increasingly used in the practice of medicine. The design of contemporary rehabilitation equipment is being constantly developed and improved. The use of automated rehabilitation stations is particularly necessary in places where they can assist rehabilitants at work. One of the areas where the application of medical devices brings significant benefits is in learning to walk again, i.e., walking re-education.

Walking re-education was originally a process used by physiotherapists who supported and protected the patient (rehabilitant). This process required a high involvement on the part of the rehabilitants, both due to the fact that the exercises were time-consuming and due to the heavy physical loads that were used. In order to improve matters, work aimed at designing rehabilitation equipment that would support the patient began [1]. The devices used in gait re-education may generally be divided into two basic types “wearables” and “non-wearables” [2].

The group of “wearable” devices includes mainly exoskeletons. One should bear in mind that exoskeletons were originally mainly used in industry and are still used in innovative industrial applications [3]. Since that time, continuous research on the development of exoskeletons has been conducted and the market offers devices driven by electric [4], hydraulic [5] and pneumatic drive systems [6]. Examples of exoskeletons that are applied for rehabilitation purposes include the HAL, ALEX, SCUT, ReWalk, or EKSO GT systems, that can be found in a review paper [7,8]. These devices are based on advanced, hybrid control algorithms that control the posture of the users and support them with the appropriate force [9], considering the interactions between the human and the robot [10]. Development work in the field of design, modelling and control of exoskeletons is still carried out and interesting examples are the results of the works [11,12,13,14].

In recent years, more and more research results have confirmed the effectiveness of the use of exoskeletons in rehabilitation practice [15]. Guidelines and efficient exercise programs have also been developed [16,17]. An important advantage of exoskeletons is the fact that they stimulate the patients to move in compliance with the correct gait patterns and force them to be active [18]. However, one should remember that rehabilitation with the use of exoskeletons cannot be used in all patients with gait disturbances. Due to the fact that the actions of the exoskeleton are initiated and controlled by the patient, a certain degree of functioning of the lower limbs and/or exteroceptive sensation are required [19]. Exoskeletons cannot be used in patients with large deformations of the lower limbs, which make it impossible to adapt the brace to the body (although persons after partial amputations and those who use prosthetics may exercise with the use of some exoskeletons). Other counterindications include, among others, obesity of the patient, unstable skeletal system, advanced osteoporosis, high spasticity or significant deformations of the spine, e.g., with a dysplasia of bones or cartilage [20,21].

An alternative approach consists in the application of “non-wearable” systems, which include both mobile devices (such as walkers and other assistive devices), as well as stationary systems equipped with a body weight support system (BWS). Relieving devices can also work with the exoskeleton, as well as with the treadmill.

Initially, the BWS systems operated like mechanical systems. This group may be divided into static body weight support systems and passive systems with counterweight, pulleys or springs. The newest gait rehabilitation support systems act dynamically, with the use of mechatronic body weight support systems [22]. Dynamic compensation of the patient’s body weight enables support with a constant vertical force, regardless of the length of the rope and the position of the person who is exercising. This type of solution enables the patient to have a more natural gait while exercising. Dynamic relief systems also enable exercises that involve kneeling, sitting or stepping on obstacles [23]. In the initial phase of rehabilitation, the use of equipment with BWS requires the participation of a physiotherapist, who verifies the correctness of the learned gait pattern. As opposed to exoskeletons, exercising with the use of body weight support systems only may result in deviations from the correct way of movement [24]. Nevertheless, BWS systems provide great assistance to therapists and are commonly applied due to their universal purpose [25]. They may be both introduced in the initial phase of re-learning to walk and used to improve the pace of walking in persons who are able to move on their own. They can both be implemented in the initial phase of the gait re-education process and used to improve speed in people who move independently [26].

The ZeroG rehabilitation device is one of the first mechatronic body weight support system, launched onto the American market in [27,28], for the specific purpose of creating an effective walking re-education process, addressing the expectations of patients and rehabilitants. It is a technologically advanced BWS system mounted on a rail, which, in turn, is attached to the ceiling. The patient who exercises with this kind of device may walk normally, but only along a route corresponding to the shape of the rail. As a result, the physiotherapist may access the rehabilitating patient at all times and will only need to focus on improving the precision of their workout since the BWS works in a fully automated way. Recently, another version of this device has been launched onto the market. An alternative option for ZeroG may be a more compact system equipped with a treadmill; such devices were developed, amongst others, at the Tarbiat Modares University in Tehran [29] and at the Silesian University of Technology in Gliwice [30]. Other devices include those equipped with orthoses to support the movement of the lower limbs, e.g., the locomat [31,32] and ReoAmbulator [33], or having programmable foot-plate-bases, such as the G-EO system [34], that enforce the movement of lower limbs while walking [35]. Several other examples of devices used in walking re-education were presented in review studies [36,37,38].

The key components of the rehabilitation equipment presented above are the BWS systems. Dynamic systems are often equipped with a series elastic actuator (SEA) drive [39,40]. The concept of this solution (Figure 1) assumes that a susceptible element that will transmit an external load is introduced to the drive system.

This type of actuator is mainly designated for mechanisms that require high accuracy of force control at low mechanical impedance [41]. As a result, such mechanisms are widely used in exoskeletons and in the BWS systems of the various rehabilitation devices described above [42,43]. The series elastic actuator drive has one power driven element and the equations of motion contain two generalized co-ordinates, so it is classified as an underactuated mechanical system (UMS) [44,45,46].

UMSs are becoming increasingly popular in various sectors of industry. These solutions are used in manipulators, vehicles, or humanoid robots. The difficulty in controlling a UMS results from the fact that techniques developed for fully actuated systems cannot be applied here [47]. These systems are not subject to feedback linearization, but they possess nonholonomic constraints and non-minimum phase characteristics [48]. Developing stable algorithms to control this class of systems is an interesting and continuously developing engineering problem.

A literature review reveals that an effective solution in UMS control is the application of a sliding mode control with various modifications [49,50,51]. This method is often used due to its high accuracy and resistance to internal and external interference. The approach based on a sliding mode control consists of two stages. The first of these is the selection of variables in the state space, determining the area of sliding movement. The second one is designing a non-continuous feedback that may force the system to achieve a state on the defined sliding movement area in a finite time [52]. The issues related to the stability of these control systems were discussed in the studies [53,54]. However, a disadvantage of the sliding mode control is the occurrence of a vibration effect that is caused by the frequency of switching the control [55]. As a result of this, the sliding mode control is not used to control the functionality of BWS systems in rehabilitation equipment. Systems used to support the patient in accommodating loads are usually controlled with the use of proven methods, in particular PD, PI, or PID regulators. However, taking into account that active body weights support systems are underactuated mechanical systems, it seems beneficial to expand the standard feedback loop to a system with compensation, or an adaptive system. These approaches are commonly used for this class of devices [56,57,58,59]. Similar and interesting example of control the BWS systems is also the use of force sensorless admittance control [60].

However, the literature review shows that the stability of BWS systems has not been analyzed enough [61] and that they have only been verified experimentally.

This paper presents two concepts for controlling BWS systems and demonstrates their stability with the use of the dynamic model and Lyapunov’s theory of stability. First, the authors present a control algorithm based on a mathematical model of the controlled object that provides a general basis for control with compensation from system dynamics. Next, an adaptive control algorithm is presented, whose application is justified by the mechanical properties of the controlled object. The object of research is a BWS system as a component of a mechatronic walking re-education device [62]. One of the key elements of this device is a polyester rope that connects the load between the system and the rehabilitant. The mathematical description of synthetic ropes is a highly complex problem and the existing modeling approaches to this issue require experimental testing [63]. Complex numerical modeling of ropes taking into account the individual fibers of the structure using the finite element method, similar to modeling composite materials, is also often used [64,65,66,67,68,69]. However, this approach makes it impossible to analyze the stability of the control system. As the parameters of synthetic ropes vary in time, the accuracy of rope models at a high number of loads is difficult to estimate [70,71]. Another problem is modeling the rope in a system where it passes through several pulleys. In order to simplify the model, the authors applied a widely used modeling process. This model omits the friction and sliding of the rope on the pulleys, does not take into account the differences in the tension of the rope on individual sections between pulleys and assumes linear characteristics of the rope, regardless of the number of load cycles. Adaptive systems that ensure good quality of control in spite of lack of knowledge or uncertainty of system parameters are well suited to such situations [72,73]. Therefore, the authors of the study developed and implemented an adaptive control algorithm, whose stability was demonstrated using Lyapunov’s theory of stability. The functionality of the proposed control algorithms was tested and compared in simulation tests.

## 2. Physical Model of the Body Weight Support System

The main task of the BWS system is to maintain a constant supporting force for the whole duration of the exercise, including the upright standing of patients who use wheelchairs. Performing certain actions can involve significant vertical displacements of the patient’s torso. During walking exercises these movements are not significant, but they may frequently occur. In order to ensure appropriate dynamics of the BWS system and a sufficient range of the vertical movement of the patient, patient relief systems are commonly equipped with two independent drive units: a rope drum drive and a drive that is responsible for minimizing the offset of the support force. A similar solution was used in the mechatronic walking re-education equipment that was tested for the purposes of this article. The BWS system is shown in Figure 2 and its detailed structure was presented in [30].

The BWS system is mostly used for maintaining a predefined support force while the patient is walking on a treadmill. Therefore, the main problem set by the authors was to verify the stability of algorithms used to control the operation of the BWS system during walking exercises. It should be noted that only one drive of the device is used during walking training, i.e., the series elastic actuator drive. The rope drum is activated generally when the patient is standing upright, or during exercises that involve significant vertical movement (squats, kneeling down, etc.). Thus, for the purposes of the research presented here, the rope drum drive was assumed to be inactive. The elements of the BWS system that play an important role during walking exercises are shown in Figure 3. In this drive, the servo motor (1) was connected to the propeller (2) with a pitch of h = 5 mm, which forces the first trolley to move (3) in the dynamic patient weight compensation system. One of the pulleys, which has a rope passing through it, was installed on the second trolley (5). Trolleys (3) and (5) are separated with springs (4) and can move linearly along the guiding rails (6). Hence, the movement of the second trolley depends on the sum of the forces acting on it as a result of the impact of springs and tension in the rope. The orthopedic harness that the patient wears is attached to one end of the rope. The other end is mounted in the rope drum, which is not shown in Figure 3.

Therefore, the system used to support a patients’ body weight during walking exercises has three degrees of freedom. The physical model, together with the generalized coordinates used, is shown in Figure 4.

In this mathematical model of the system analyzed, two generalized coordinates are used, which are connected with the rotation angle of the rotor in the engine, $\varphi_{Z}$ and with the displacement of the pulley installed in the compensating system $x_{w}$. Shifting the rope installation point to the patient’s harness constitutes kinematic input.

## 3. Equations of Motion of the Body Weight Support System

The dynamic properties of the system are defined using differential equations of motion in the following form:(1)$$\{\begin{array}{c} I_{Z}{\ddot{\varphi }}_{Z}+b_{Z}{\dot{\varphi }}_{Z}+b_{s}\left({\dot{\varphi }}_{Z}\frac{h^{2}}{4\pi^{2}}-{\dot{x}}_{w}\frac{h}{2\pi }\right)+k_{s}\left(\varphi_{Z}\frac{h^{2}}{4\pi^{2}}-x_{w}\frac{h}{2\pi }\right)=M_{Z}\\ m_{w}{\ddot{x}}_{w}-b_{s}\left({\dot{\varphi }}_{Z}\frac{h}{2\pi }-{\dot{x}}_{w}\right)-k_{s}\left(\varphi_{Z}\frac{h}{2\pi }-x_{w}\right)+2k_{l}\left(2x_{w}-z_{p}\right)=0 \end{array}$$
where $\varphi_{Z}$ is the angular rotation of the drive motor shaft, $x_{w}$ is the displacement of the passive part of the system, $I_{Z}$ is the reduced mass moment of inertia of the active part of the system, $m_{Z}$ is the reduced mass of the passive part of the system, $b_{Z}$ is the reduced resistance to the motion coefficient of the active part of the system, $b_{s}$ is the attenuation coefficient of the shock-absorbing system, $k_{s}$ is the modulus of elasticity of the shock-absorbing springs, $k_{l}$ is the modulus of elasticity of the rope, h is the pitch of the ball screw thread in the drive system and $M_{Z}$ is the drive torque.

The equations of motion of the system are defined in form of the matrix:(2)$$\begin{array}{c} \left[\begin{array}{cc} I_{Z} & 0\\ 0 & m_{w} \end{array}\right]\left[\begin{array}{c} {\ddot{\varphi }}_{Z}\\ {\ddot{x}}_{w} \end{array}\right]+\left[\begin{array}{cc} b_{Z}+b_{s}\frac{h^{2}}{4\pi^{2}} & -b_{s}\frac{h}{2\pi }\\ -b_{s}\frac{h}{2\pi } & b_{s} \end{array}\right]\left[\begin{array}{c} {\dot{\varphi }}_{Z}\\ {\dot{x}}_{w} \end{array}\right]+\left[\begin{array}{cc} k_{s}\frac{h^{2}}{4\pi^{2}} & -k_{s}\frac{h}{2\pi }\\ -k_{s}\frac{h}{2\pi } & k_{s}+4k_{l} \end{array}\right]\left[\begin{array}{c} \varphi_{Z}\\ x_{w} \end{array}\right]+\\ +\left[\begin{array}{c} 0\\ -2k_{l} \end{array}\right]z_{p}=\left[\begin{array}{c} M_{Z}\\ 0 \end{array}\right] \end{array}$$
using the following definitions:(3)$$M=\left[\begin{array}{cc} I_{Z} & 0\\ 0 & m_{w} \end{array}\right]$$
(4)$$B=\left[\begin{array}{cc} b_{Z}+b_{s}\frac{h^{2}}{4\pi^{2}} & -b_{s}\frac{h}{2\pi }\\ -b_{s}\frac{h}{2\pi } & b_{s} \end{array}\right],$$
(5)$$K=\left[\begin{array}{cc} k_{s}\frac{h^{2}}{4\pi^{2}} & -k_{s}\frac{h}{2\pi }\\ -k_{s}\frac{h}{2\pi } & k_{s}+4k_{l} \end{array}\right],$$
(6)$$q=\left[\begin{array}{c} \varphi_{Z}\\ x_{w} \end{array}\right]=\left[\begin{array}{c} q_{1}\\ q_{2} \end{array}\right],$$
(7)$$U=\left[\begin{array}{c} M_{Z}\\ 0 \end{array}\right],$$
(8)$$\xi \left(t\right)=\left[\begin{array}{c} 0\\ -2k_{l} \end{array}\right],$$

The equations of motion are then formulated in a more compact form:(9)$$M\ddot{q}+B\dot{q}+Kq+\xi \left(t\right)z_{p}=U$$
where M is the matrix of inertia, B is the matrix of attenuation coefficients, K is the matrix of elasticity modules, q is the vector of generalized coordinates, U is the control vector and ξ(t) is the disturbance vector.

The Equations of motion (1) contain an expression that describes the reaction force of the elastic rope:(10)$$F_{l}=k_{l}\left(2x_{w}-z_{p}\right)$$

This reaction force is the value that is to be calculated by using the system. Its determination, based on the co-ordinate $x_{w}$, requires precise knowledge of the rope elasticity modulus, $k_{l}$. Thus, the variables are transformed, so that $x_{w}$, the coordinate from the equations of motion, is eliminated and the description of dynamic properties of the system is expressed as a function of force, $F_{l}$.

Rope deformation is denoted as Δ leading to the following equations:(11)$$F_{l}=k_{l}\Delta$$
where
(12)$$\Delta =2x_{w}-z_{p}$$

Next, Equation (12) is multiplied by $k_{l}$, giving
(13)$$F_{l}=2k_{l}x_{w}-k_{l}z_{p}$$

In order to transform the description of the dynamic properties of the system into different variables, a new vector of coordinates is used and is referred to as the vector of task coordinates:(14)$$\theta =\left[\begin{array}{c} \varphi_{Z}\\ F_{l} \end{array}\right]$$

It is assumed that all elements of Vector (14) are available for measurement. The following relation exists between the coordinates:(15)$$\left[\begin{array}{c} \varphi_{Z}\\ F_{l} \end{array}\right]=\left[\begin{array}{cc} 1 & 0\\ 0 & 2k_{l} \end{array}\right]\left[\begin{array}{c} \varphi_{Z}\\ x_{w} \end{array}\right]+\left[\begin{array}{c} 0\\ -k_{l} \end{array}\right]z_{p}$$

In a compact matrix form, it may be presented as the relation between vectors θ and q in the following way:(16)$$\theta =T_{1}q+T_{2}z_{p}$$
where
(17)$$T_{1}=\left[\begin{array}{cc} 1 & 0\\ 0 & 2k_{l} \end{array}\right]$$
(18)$$T_{2}=\left[\begin{array}{c} 0\\ -k_{l} \end{array}\right].$$

The vector of generalized coordinates is determined from Equation (16):(19)$$q=T_{1}^{-1}\theta -T_{1}^{-1}T_{2}z_{p}$$

The notation
(20)$$E=T_{1}^{-1}=\left[\begin{array}{cc} 1 & 0\\ 0 & \frac{1}{2K_{l}} \end{array}\right]$$
is used to write Equation (19) in the form
(21)$$q=E\theta -ET_{2}z_{p}$$

The derivatives of this equation are
(22)$$\dot{q}=E\dot{\theta }-ET_{2}{\dot{z}}_{p}$$
(23)$$\ddot{q}=E\ddot{\theta }-ET_{2}{\ddot{z}}_{p}.$$

Including Equations (21)–(23) in Equation (9), equations of motion were obtained and expressed in terms of task coordinates:(24)$$M\left(E\ddot{\theta }-ET_{2}{\ddot{z}}_{p}\right)+B\left(E\dot{\theta }-ET_{2}{\dot{z}}_{p}\right)+K\left(E\theta -ET_{2}z_{p}\right)+\xi \left(t\right)z_{p}=U$$

Ordering the equation results in
(25)$$ME\ddot{\theta }+BE\dot{\theta }+KE\theta +\xi \left(t\right)z_{p}-\left(MET_{2}{\ddot{z}}_{p}+BET_{2}{\dot{z}}_{p}+KET_{2}z_{p}\right)=U$$

With this approach, the description of the dynamic properties is presented in the form of a function comprising of the co-ordinate that describes the movement of the active part of the system and tension force of the rope. However, the variable describing the movement of the passive part of the system was eliminated, since it influences the tension force of the rope, but it is not the main object of interest in this paper. Equation (25) was presented in a more concise form:(26)$$M_{1}\ddot{\theta }+B_{1}\dot{\theta }+K_{1}\theta -\left(m_{2}{\ddot{z}}_{p}+b_{2}{\dot{z}}_{p}+k_{2}z_{p}\right)=U$$
where
(27)$$M_{1}=ME=\left[\begin{array}{cc} I_{Z} & 0\\ 0 & \frac{1}{2}\frac{m_{w}}{k_{l}} \end{array}\right]=\left[\begin{array}{cc} a_{1} & 0\\ 0 & \frac{1}{2}a_{2} \end{array}\right]$$
(28)$$B_{1}=BE=\left[\begin{array}{cc} b_{Z}+b_{s}\frac{h^{2}}{4\pi^{2}} & -\frac{1}{2}\frac{b_{s}}{k_{l}}\frac{h}{2\pi }\\ -b_{s}\frac{h}{2\pi } & b_{s}\frac{1}{2k_{l}} \end{array}\right]=\left[\begin{array}{cc} a_{3}+a_{4}\frac{h^{2}}{4\pi^{2}} & -\frac{1}{2}a_{5}\frac{h}{2\pi }\\ -a_{4}\frac{h}{2\pi } & \frac{1}{2}a_{5} \end{array}\right],$$
(29)$$K_{1}=KE=\left[\begin{array}{cc} k_{s}\frac{h^{2}}{4\pi^{2}} & -\frac{1}{2}\frac{k_{s}}{k_{l}}\frac{h}{2\pi }\\ -k_{s}\frac{h}{2\pi } & \frac{1}{2}\frac{k_{s}}{k_{l}}+2 \end{array}\right]=\left[\begin{array}{cc} a_{6}\frac{h^{2}}{4\pi^{2}} & -\frac{1}{2}a_{7}\frac{h}{2\pi }\\ -a_{6}\frac{h}{2\pi } & \frac{1}{2}a_{7}+2 \end{array}\right],$$
(30)$$m_{2}=MET_{2}=\left[\begin{array}{cc} I_{Z} & 0\\ 0 & m_{w} \end{array}\right]\left[\begin{array}{c} 0\\ -\frac{1}{2} \end{array}\right]=\left[\begin{array}{c} 0\\ -\frac{1}{2}a_{8} \end{array}\right],$$
(31)$$b_{2}=BET_{2}=\left[\begin{array}{cc} b_{Z}+b_{s}\frac{h^{2}}{4\pi^{2}} & -b_{s}\frac{h}{2\pi }\\ -b_{s}\frac{h}{2\pi } & b_{s} \end{array}\right]\left[\begin{array}{c} 0\\ -\frac{1}{2} \end{array}\right]=\left[\begin{array}{c} \frac{1}{2}a_{4}\frac{h}{2\pi }\\ -\frac{1}{2}a_{4} \end{array}\right],$$
(32)$$k_{2}=KET_{2}-\xi \left(t\right)=\left[\begin{array}{cc} k_{s}\frac{h^{2}}{4\pi^{2}} & -k_{s}\frac{h}{2\pi }\\ -k_{s}\frac{h}{2\pi } & k_{s}+4k_{l} \end{array}\right]\left[\begin{array}{c} 0\\ -\frac{1}{2} \end{array}\right]-\left[\begin{array}{c} 0\\ -2k_{l} \end{array}\right]=\left[\begin{array}{c} \frac{1}{2}a_{6}\frac{h}{2\pi }\\ -\frac{1}{2}a_{6} \end{array}\right],$$
and the physical description of the parameters is as follows:(33)$$\{\begin{array}{c} a_{1}=I_{Z}\\ a_{2}=\frac{m_{w}}{k_{l}}\\ a_{3}=b_{Z}\\ a_{4}=b_{s}\\ a_{5}=\frac{b_{s}}{k_{l}}\\ a_{6}=k_{s}\\ a_{7}=\frac{k_{s}}{k_{l}}\\ a_{8}=m_{w} \end{array}.$$

It is assumed that the parameters are the ranges of a constant. The formula $\left(m_{2}{\ddot{z}}_{p}+b_{2}{\dot{z}}_{p}+k_{2}z_{p}\right)$ corresponds to part of the system dynamics that is stimulated by the movement of the end of the rope. Knowledge of the parameters of the movement of the end of the rope enables us to take the formula into account, otherwise it may be treated as interference.

It should be noted that force $F_{l}$ is not controlled directly and that it is the variable that corresponds to the passive part of the system. The active, i.e., controlled, variable is $\varphi_{Z}$. The system described belongs to the UMS class. As a result, the vector of task coordinates was decomposed as follows:(34)$$\theta =\left[\begin{array}{c} \theta_{a}\\ \theta_{p} \end{array}\right]$$
where $\theta_{a}$ is the vector of active coordinates and $\theta_{p}$ is the vector of passive coordinates. In the case analyzed, these vectors take the following form:(35)$$\theta_{a}=\left[\varphi_{Z}\right]$$
(36)$$\theta_{p}=\left[F_{l}\right].$$

The matrix and vectors in Equation (26) are also decomposed in the following way:(37)$$M_{1}=\left[\begin{array}{cc} M_{1aa} & M_{1ap}\\ M_{1pa} & M_{1pp} \end{array}\right]$$
(38)$$B_{1}=\left[\begin{array}{cc} B_{1aa} & B_{1ap}\\ B_{1pa} & B_{1pp} \end{array}\right],$$
(39)$$K_{1}=\left[\begin{array}{cc} K_{1aa} & K_{1ap}\\ K_{1pa} & K_{1pp} \end{array}\right],$$
(40)$$m_{2}=\left[\begin{array}{c} m_{2a}\\ m_{2p} \end{array}\right],$$
(41)$$b_{2}=\left[\begin{array}{c} b_{2a}\\ b_{2p} \end{array}\right],$$
(42)$$k_{2}=\left[\begin{array}{c} k_{2a}\\ k_{2p} \end{array}\right],$$
(43)$$U=\left[\begin{array}{c} u\\ 0 \end{array}\right].$$

As a result, Equation (26) can be written in the form
(44)$$\left[\begin{array}{cc} M_{1aa} & M_{1ap}\\ M_{1pa} & M_{1pp} \end{array}\right]\left[\begin{array}{c} {\ddot{\theta }}_{a}\\ {\ddot{\theta }}_{p} \end{array}\right]+\left[\begin{array}{c} N_{a}\\ N_{p} \end{array}\right]=\left[\begin{array}{c} u\\ 0 \end{array}\right],$$
where
(45)$$N_{a}=B_{1aa}{\dot{\theta }}_{a}+B_{1ap}{\dot{\theta }}_{p}+K_{1aa}\theta_{a}+K_{1ap}\theta_{p}-m_{2a}{\ddot{z}}_{p}-b_{2a}{\dot{z}}_{p}-k_{2a}z_{p}$$
(46)$$N_{p}=B_{1pa}{\dot{\theta }}_{a}+B_{1pp}{\dot{\theta }}_{p}+K_{1pa}\theta_{a}+K_{1pp}\theta_{p}-m_{2p}{\ddot{z}}_{p}-b_{2p}{\dot{z}}_{p}-k_{2p}z_{p}$$

Equations (26) and (44) will be used later in this article in the synthesis of system control.

## 4. Tracking Movement

In order to provide patient relief by using the device described in Section 2, it is necessary to calculate the force supporting the patient. Thus, the task of the control system is to track the movement of the trajectory. The task of performing tracking movements can be replaced with the task of stabilizing the error around the trajectory; this is a simpler issue from a mathematical point of view.

The trajectory set of the passive part of the system is defined as $\theta_{pd}\left(t\right)$, which has continuous derivatives. It is also assumed that a trajectory exists for the active part of the system, $\theta_{ad}\left(t\right)$, having continuous derivatives, i.e.,
(47)$$\theta_{d}=\left[\begin{array}{c} \theta_{ad}\\ \theta_{pd} \end{array}\right]$$
(48)$${\dot{\theta }}_{d}=\left[\begin{array}{c} {\dot{\theta }}_{ad}\\ {\dot{\theta }}_{pd} \end{array}\right],$$
(49)$${\ddot{\theta }}_{d}=\left[\begin{array}{c} {\ddot{\theta }}_{ad}\\ {\ddot{\theta }}_{pd} \end{array}\right].$$

The error in the calculation of the trajectory is defined as
(50)$$e=\theta_{d}-\theta$$

It may also be expressed in the form $e={\left[\begin{array}{cc} e_{a}^{T} & e_{p}^{T} \end{array}\right]}^{T}$, where $e_{a}$ and $e_{p}$ are the tracking errors of the active and passive parts of the system, respectively. The generalized tracking error was used:(51)$$s=\dot{e}+\Lambda e$$

This may be expressed as $s={\left[\begin{array}{cc} s_{a}^{T} & s_{p}^{T} \end{array}\right]}^{T}$, where $s_{a}$ and $s_{p}$ are the generalized tracking errors of the active and passive parts of the system, respectively, and Λ is the matrix, where $\Lambda =\Lambda^{T}>0$. Matrix Λ may be written as follows: $\Lambda =\left[\begin{array}{cc} \Lambda_{a} & 0\\ 0 & \Lambda_{p} \end{array}\right]$, where $\Lambda_{a}=\Lambda_{a}^{T}>0$ i $\Lambda_{p}=\Lambda_{p}^{T}>0$.

Equation (51) can be broken down into further expressions:(52)$$s={\dot{\theta }}_{d}-\dot{\theta }+\Lambda e$$
(53)$$\dot{\theta }=-s+{\dot{\theta }}_{d}+\Lambda e,$$
(54)$$\ddot{\theta }=-\dot{s}+{\ddot{\theta }}_{d}+\Lambda \dot{e}.$$

Equation (54) is included in Equation (26), resulting in the definition
(55)$$M_{1}\left(-\dot{s}+{\ddot{\theta }}_{d}+\Lambda \dot{e}\right)+B_{1}\dot{\theta }+K_{1}\theta -\left(m_{2}{\ddot{z}}_{p}+b_{2}{\dot{z}}_{p}+k_{2}z_{p}\right)=u$$

An auxiliary variable is defined as
(56)$$v={\dot{\theta }}_{d}+\Lambda e$$
and this is used in the definition of part of the system dynamics:(57)$$f=M_{1}\dot{v}+B_{1}\dot{\theta }+K_{1}\theta -\left(m_{2}{\ddot{z}}_{p}+b_{2}{\dot{z}}_{p}+k_{2}z_{p}\right)$$
resulting in
(58)$$M_{1}\dot{s}=-U+f$$

Equation (58) describes the dynamic properties of the system as a function of the generalized tracking error. In this way, the problem of tracking the trajectory was transformed by the task of minimizing the tracking error.

## 5. Control Algorithm Based on the Mathematical Model

This section presents the control algorithm of the basic system, with the control law taking into account the dynamic properties described by the mathematical model. PD control was assumed, as well as perfect compensation control resulting from the description of system dynamics, i.e.,
(59)$$U=f+K_{D}s$$
where the ***f*** function is given by Equation (57) and PD control has the form
(60)$$K_{D}s=K_{D}\dot{e}+K_{D}\Lambda e$$
where $K_{D}=K_{D}^{T}>0$. The $K_{D}$ matrix is a differential reinforcement matrix and $K_{D}\Lambda$ is a proportional reinforcement matrix. The $K_{D}$ matrix may be expressed as follows:(61)$$K_{D}=\left[\begin{array}{cc} K_{Da} & 0\\ 0 & K_{Dp} \end{array}\right]$$
where $K_{Da}=K_{Da}^{T}>0$ and $K_{Dp}=K_{Dp}^{T}>0$.

At this stage, it was assumed that the f function is known, which means that the interference $\left(m_{2}{\ddot{z}}_{p}+b_{2}{\dot{z}}_{p}+k_{2}z_{p}\right)$ is also known. This requires the ability to measure the movement of the end of the rope generated by the patient. The f function given by Equation (57) was decomposed in the following way:(62)$$f=\left[\begin{array}{c} f_{a}\\ f_{p} \end{array}\right]$$
where
(63)$$\begin{array}{c} f_{a}=M_{1aa}{\dot{v}}_{a}+M_{1ap}{\dot{v}}_{p}+B_{1aa}{\dot{\theta }}_{a}+B_{1ap}{\dot{\theta }}_{p}+K_{1aa}\theta_{a}+K_{1ap}\theta_{p}-m_{2a}{\ddot{z}}_{p}-b_{2a}{\dot{z}}_{p}-k_{2a}z_{p}\\ =M_{1aa}{\dot{v}}_{a}+M_{1ap}{\dot{v}}_{p}+N_{a}, \end{array}$$
(64)$$\begin{array}{c} f_{p}=M_{1pa}{\dot{v}}_{a}+M_{1pp}{\dot{v}}_{p}+B_{1pa}{\dot{\theta }}_{a}+B_{1pp}{\dot{\theta }}_{p}+K_{1pa}\theta_{a}+K_{1pp}\theta_{p}-m_{2p}{\ddot{z}}_{p}-b_{2p}{\dot{z}}_{p}-k_{2p}z_{p}\\ =M_{1pa}{\dot{v}}_{a}+M_{1pp}{\dot{v}}_{p}+N_{p}. \end{array}$$
where vectors ${\dot{v}}_{a}$ and ${\dot{v}}_{p}$ are elements of vector $\dot{v}$, decomposed as $\dot{v}={\left[\begin{array}{cc} {\dot{v}}_{a}^{T} & {\dot{v}}_{p}^{T} \end{array}\right]}^{T}$.

Based on the decompositions of the vectors and matrices shown, the control (59) was expressed in the following form:(65)$$U=\left[\begin{array}{c} u\\ 0 \end{array}\right]=\left[\begin{array}{c} f_{a}\\ f_{p} \end{array}\right]+\left[\begin{array}{c} K_{Da}s_{a}\\ K_{Dp}s_{p} \end{array}\right]$$
giving
(66)$$u=f_{a}+K_{Da}s_{a}$$
(67)$$0=f_{p}+K_{Dp}s_{p}.$$

The control vector U, given by Equation (65), has two component vectors; however, the second one, defined by Equation (67), is equal to zero. It is referred to as fictitious control, since it does not directly influence the movement of the passive part of the system. However, it describes the dynamic properties of part of the system, as it contains the $f_{p}$ function. This equation may be used to determine the trajectory of the active part of the system, whose calculation will ensure desirable behavior of the passive part, and, ultimately, the calculation of the trajectory $\theta_{pd}$. The detailed form of this equation is
(68)$$M_{1pa}{\dot{v}}_{a}+M_{1pp}{\dot{v}}_{p}+B_{1pa}{\dot{\theta }}_{a}+B_{1pp}{\dot{\theta }}_{p}+K_{1pa}\theta_{a}+K_{1pp}\theta_{p}-m_{2p}{\ddot{z}}_{p}-b_{2p}{\dot{z}}_{p}-k_{2p}z_{p}+K_{Dp}s_{p}=0$$

The vector of the velocity of the active part of the system was determined by Equation (68):(69)$${\dot{\theta }}_{a}=-B_{1pa}^{-1}\left[M_{1pa}{\dot{v}}_{a}+M_{1pp}{\dot{v}}_{p}+B_{1pp}{\dot{\theta }}_{p}+K_{1pa}\theta_{a}+K_{1pp}\theta_{p}-m_{2p}{\ddot{z}}_{p}-b_{2p}{\dot{z}}_{p}-k_{2p}z_{p}+K_{Dp}s_{p}\right]$$

It was assumed that matrix $B_{1pa}$ is invertible. In the event that it is not a square matrix, the Moore–Penrose pseudoinverse should be applied, $B_{1pa}^{+}$ instead of $B_{1pa}^{-1}$ [74].

The velocity determined by Equation (69) was adopted as the set velocity of the active part of the system:(70)$${\dot{\theta }}_{ad}=-B_{1pa}^{-1}\left[M_{1pa}{\dot{v}}_{a}+M_{1pp}{\dot{v}}_{p}+B_{1pp}{\dot{\theta }}_{p}+K_{1pa}\theta_{a}+K_{1pp}\theta_{p}-m_{2p}{\ddot{z}}_{p}-b_{2p}{\dot{z}}_{p}-k_{2p}z_{p}+K_{Dp}s_{p}\right]$$

Then, by integration and differentiation, the following is determined:(71)$$\theta_{ad}=\int_{0}^{t}{\dot{\theta }}_{ad}dt,$$
(72)$${\ddot{\theta }}_{ad}=\frac{d}{dt}{\dot{\theta }}_{ad}.$$

Equations (70)–(72) give the trajectory of the active part of the system and the purpose of the control system is to calculate this. The calculation of this trajectory will ensure the calculation of the trajectory of the passive part of the system, i.e., achieving the desirable tension force of the rope.

### 5.1. Stability of a Closed-Loop System

The stability of the closed-loop system, i.e., considering control, was demonstrated with the use of Lyapunov’s theory of stability. Therefore, Lyapunov’s function was used as a square form of the generalized tracking error:(73)$$L=\frac{1}{2}s^{T}M_{1}s$$

The derivative of this function in time equals
(74)$$\dot{L}=s^{T}M_{1}\dot{s}+\frac{1}{2}s^{T}{\dot{M}}_{1}s$$

The matrix of inertia $M_{1}$ is constant, hence ${\dot{M}}_{1}=0$, and the second part of function (74) equals zero. Considering the description of system dynamics in the form of Equation (58), the derivative of function *L* was written on system trajectories as
(75)$$\dot{L}=s^{T}\;\left(-U+f\right)$$

Considering the form of vectors s, U and f**,** it was written
(76)$$\dot{L}=s^{T}\left[\begin{array}{c} -u+f_{a}\\ f_{p} \end{array}\right]=s^{T}\left[\begin{array}{c} -K_{Da}s_{a}\\ f_{p} \end{array}\right]$$
using Equation (66). Function $f_{p}$, determined by Equation (64), using Equation (69), gives
(77)$$f_{p}=-K_{Dp}s_{p}$$

Based on that, function (76) can now be written as
(78)$$\dot{L}=s^{T}\left[\begin{array}{c} -K_{Da}s_{a}\\ -K_{Dp}s_{p} \end{array}\right]=s^{T}K_{D}s\leq 0$$

As function L is positively defined in s space and the analysis performed demonstrates that the derivative $\dot{L}$ is negatively semi-definite, according to Lyapunov’s theory of stability, the variable s is limited. The application of Barbalat’s lemma allows us to demonstrate that s converges to zero as a result of the convergence of error e to zero.

### 5.2. Discussion of Results

This approach requires setting the trajectory of the passive part of the system in time, which will constitute the basis for calculating the trajectory of the active part from Equation (70). The control law (66) enables the calculation of the trajectory of the passive part, i.e., obtaining the rope set tension force, because the dynamic properties of this part of the system have been taken into account in the trajectory of the active part. The calculation of the adopted control law requires knowledge of the parameters of the model (inertia, attenuation and elasticity), the state of the system (displacements and velocities) and interference.

## 6. Adaptive Control System

The main disadvantage of the control algorithm described in Section 5 is the fact that it requires thorough knowledge of the parameters of the mechanical system. Inaccurate knowledge of these parameters results in inaccurate control. Due to that, in this section, the authors introduce an adaptive control algorithm based on the mathematical structure of the mechanical system model. This algorithm does not require knowledge of the parameters of the object, which are being adapted during its operation.

The assumptions are that PD control and compensation control are generated by the adaptive controller, i.e.,
(79)$$U=\hat{f}+K_{D}s$$
where PD control in the form $K_{D}s$ is described by Equation (60). The $\hat{f}$ element approximates the f function given by Equation (57). The f function was decomposed according to Equation (62) and specific elements of this equation were written in linear form to the parameters, i.e.,
(80)$$f=\left[\begin{array}{c} f_{a}\\ f_{p} \end{array}\right]=\left[\begin{array}{c} Y_{a}a\\ Y_{p}a \end{array}\right]$$
where $Y_{a}$ and $Y_{p}$ are the regression matrices and a is the vector of the model parameters.

Assuming that the parameter vector a is not exactly known, but the structure of the model is known, compensation control can be defined in the form:(81)$$\hat{f}=\left[\begin{array}{c} {\hat{f}}_{a}\\ {\hat{f}}_{p} \end{array}\right]=\left[\begin{array}{c} Y_{a}\hat{a}\\ Y_{p}\hat{a} \end{array}\right]$$
where $\hat{a}$ is the estimate of the parameter vector a.

Based on the decomposition introduced, the control (79) was written in the following form:(82)$$U=\left[\begin{array}{c} u\\ 0 \end{array}\right]=\left[\begin{array}{c} {\hat{f}}_{a}\\ {\hat{f}}_{p} \end{array}\right]+\left[\begin{array}{c} K_{Da}s_{a}\\ K_{Dp}s_{p} \end{array}\right]$$

The first part of Equation (82), given by the relationship
(83)$$u={\hat{f}}_{a}+K_{Da}s_{a}$$
is the control of the active part of the system, while the expression
(84)$${\hat{f}}_{p}+K_{Dp}s_{p}=0$$
is the fictitious control, as it does not directly influence the movement of the passive part of the system. Referring to the process described in Section 5, this equation was used to determine the trajectory of the active part of the system. The detailed form of Equation (84) is as follows:(85)$${\hat{M}}_{1pa}{\dot{v}}_{a}+{\hat{M}}_{1pp}{\dot{v}}_{p}+{\hat{B}}_{1pa}{\dot{\theta }}_{a}+{\hat{B}}_{1pp}{\dot{\theta }}_{p}+{\hat{K}}_{1pa}\theta_{a}+{\hat{K}}_{1pp}\theta_{p}-{\hat{m}}_{2p}{\ddot{z}}_{p}-{\hat{b}}_{2p}{\dot{z}}_{p}-{\hat{k}}_{2p}z_{p}+K_{Dp}s_{p}=0$$
where the ^ symbol above matrices and vectors means that they are dependent on the estimates of parameters.

The velocity vector of the active part of the system is determined from Equation (85):(86)$${\dot{\theta }}_{a}=-{\hat{B}}_{1pa}^{-1}\left[{\hat{M}}_{1pa}{\dot{v}}_{a}+{\hat{M}}_{1pp}{\dot{v}}_{p}+{\hat{B}}_{1pp}{\dot{\theta }}_{p}+{\hat{K}}_{1pa}\theta_{a}+{\hat{K}}_{1pp}\theta_{p}-{\hat{m}}_{2p}{\ddot{z}}_{p}-{\hat{b}}_{2p}{\dot{z}}_{p}-{\hat{k}}_{2p}z_{p}+K_{Dp}s_{p}\right].$$

This is based on the assumption that matrix ${\hat{B}}_{1pa}$ is invertible. When ${\hat{B}}_{1pa}$ is not a square matrix, the Moore–Penrose pseudoinverse should be applied, ${\hat{B}}_{1pa}^{+}$ instead of ${\hat{B}}_{1pa}^{-1}$ [74]. The velocity given by Equation (86) was adopted as the set velocity of the active part of the system:(87)$${\dot{\theta }}_{ad}=-{\hat{B}}_{1pa}^{-1}\left[{\hat{M}}_{1pa}{\dot{v}}_{a}+{\hat{M}}_{1pp}{\dot{v}}_{p}+{\hat{B}}_{1pp}{\dot{\theta }}_{p}+{\hat{K}}_{1pa}\theta_{a}+{\hat{K}}_{1pp}\theta_{p}-{\hat{m}}_{2p}{\ddot{z}}_{p}-{\hat{b}}_{2p}{\dot{z}}_{p}-{\hat{k}}_{2p}z_{p}+K_{Dp}s_{p}\right].$$

By integrating and differentiating, the following was determined:(88)$$\theta_{ad}=\int_{0}^{t}{\dot{\theta }}_{ad}dt$$
(89)$${\ddot{\theta }}_{ad}=\frac{d}{dt}{\dot{\theta }}_{ad}.$$

Equations (87)–(89) give the trajectory of the active part of the system, which is the purpose of the control system. This calculation will ensure the calculation of the trajectory of the passive part of the system, i.e., achieving the desirable rope tension force.

Note that the trajectory depends on the estimates of the model parameters. In the event that the adaptation of parameter estimates starts from zero, then the trajectory of the passive part of the system will be calculated with a large error in the initial phase of movement, in spite of the fact that the active part allows calculation of the trajectory. This error will diminish with the use of parameter estimates.

### 6.1. Stability of the Adaptive System

The stability of the closed-loop system was demonstrated with the use of Lyapunov’s theory of stability. Therefore, Lyapunov’s function was adopted as a square form of the generalized tracking error:(90)$$L=\frac{1}{2}s^{T}M_{1}s+\frac{1}{2}{\tilde{a}}^{T}\Gamma^{-1}\tilde{a}$$
where Γ is the matrix of adaptation reinforcement, such that $\Gamma =\Gamma^{T}>0$, e.g., $\Gamma =\mathrm{diag}\left\{\gamma_{j}\right\}$, while
(91)$$\tilde{a}=a-\hat{a}$$
is the vector of parameter estimation error. The derivative of function (91) with time equals
(92)$$\dot{L}=s^{T}M_{1}\dot{s}+{\tilde{a}}^{T}\Gamma^{-1}\dot{\tilde{a}}$$
and is based on the fact that matrices $M_{1}$ and Γ are constant, hence ${\dot{M}}_{1}=0$ and $\dot{\Gamma }=0$. Using the description of system dynamics in the form of Equation (58), the derivative of the L function was written on the system trajectories as
(93)$$\dot{L}=s^{T}\left(-U+f\right)+{\tilde{a}}^{T}\Gamma^{-1}\dot{\tilde{a}}$$

Using the control Equation (82) and the decomposed form of the *f* function, it can be written as
(94)$$\dot{L}=s^{T}\left[\begin{array}{c} -K_{Da}s_{a}+f_{a}-{\hat{f}}_{a}\\ f_{p} \end{array}\right]+{\tilde{a}}^{T}\Gamma^{-1}\dot{\tilde{a}}$$

The expression ${\hat{B}}_{1pa}{\dot{\theta }}_{a}$ was added and deducted in the matrix as below, to yield
(95)$$\dot{L}=s^{T}\left[\begin{array}{c} -K_{Da}s_{a}+f_{a}-{\hat{f}}_{a}\\ f_{p}+{\hat{B}}_{1pa}{\dot{\theta }}_{a}-{\hat{B}}_{1pa}{\dot{\theta }}_{a} \end{array}\right]+{\tilde{a}}^{T}\Gamma^{-1}\dot{\tilde{a}}$$

Using Equation (86), the expression in Equation (95) is written in the form
(96)$$\begin{array}{c} {\hat{B}}_{1pa}{\dot{\theta }}_{a}-{\hat{B}}_{1pa}{\dot{\theta }}_{a}\\ =-\left[{\hat{M}}_{1pa}{\dot{v}}_{a}+{\hat{M}}_{1pp}{\dot{v}}_{p}+{\hat{B}}_{1pa}{\dot{\theta }}_{a}+{\hat{B}}_{1pp}{\dot{\theta }}_{p}+{\hat{K}}_{1pa}\theta_{a}+{\hat{K}}_{1pp}\theta_{p}-{\hat{m}}_{2p}{\ddot{z}}_{p}-{\hat{b}}_{2p}{\dot{z}}_{p}-{\hat{k}}_{2p}z_{p}+K_{Dp}s_{p}\right]\\ =-{\hat{f}}_{p}-K_{Dp}s_{p}. \end{array}$$

Then, function (95) can be rewritten in the following form:(97)$$\begin{array}{l} \dot{L}=s^{T}\left[\begin{array}{c} -K_{Da}s_{a}+f_{a}-{\hat{f}}_{a}\\ -K_{Dp}s_{p}+f_{p}-{\hat{f}}_{p} \end{array}\right]+{\tilde{a}}^{T}\Gamma^{-1}\dot{\tilde{a}\;}=s^{T}\left[\begin{array}{c} -K_{Da}s_{a}+Y_{a}a-Y_{a}\hat{a}\\ -K_{Dp}s_{p}+Y_{p}a-Y_{p}\hat{a} \end{array}\right]+{\tilde{a}}^{T}\Gamma^{-1}\dot{\tilde{a}}\\ \;=-s^{T}\left[\begin{array}{c} K_{Da}s_{a}\\ K_{Dp}s_{p} \end{array}\right]+s^{T}\left[\begin{array}{c} Y_{a}\tilde{a}\\ Y_{p}\tilde{a} \end{array}\right]+{\tilde{a}}^{T}\Gamma^{-1}\dot{\tilde{a}\;}=-s^{T}K_{D}s+s^{T}\left[\begin{array}{c} Y_{a}\\ Y_{p} \end{array}\right]\tilde{a}+{\tilde{a}}^{T}\Gamma^{-1}\dot{\tilde{a}\;}\\ \;=-s^{T}K_{D}s+{\tilde{a}}^{T}({\left[\begin{array}{c} Y_{a}\\ Y_{p} \end{array}\right]}^{T}s+\Gamma^{-1}\dot{\tilde{a}\;}). \end{array}$$

The law of model parameter adaptation is used in the form of a differential equation:(98)$$\dot{\hat{a}}=\Gamma {\left[\begin{array}{c} Y_{a}\\ Y_{p} \end{array}\right]}^{T}s$$

Putting Equation (98) into Equation (97), the final result is obtained:(99)$$\dot{L}=-s^{T}K_{D}s+{\tilde{a}}^{T}\left({\left[\begin{array}{c} Y_{a}\\ Y_{p} \end{array}\right]}^{T}s-\Gamma^{-1}\Gamma {\left[\begin{array}{c} Y_{a}\\ Y_{p} \end{array}\right]}^{T}s\right)=-s^{T}K_{D}s\leq 0$$

As the function L is positively defined in the space of variable s and the conducted analysis demonstrates that the derivative $\dot{L}$ is negatively semi-definite, according to Lyapunov’s theory of stability, the variable s is limited. The application of Barbalat’s lemma allows us to demonstrate that variable s is convergent to zero as a result of the convergence of error e to zero. The vector of parameter estimation error $\tilde{a}$ is limited, but it was not proven that it converges to zero. However, this is not a problem for this work, since it is not the aim of this paper to control for parameter estimates to converge to actual parameters. It is acceptable if these parameters are limited.

### 6.2. Discussion of Results

This approach to the problem means that model parameter estimates are subject to change during the operation of the system. Tracking errors change the most in the initial phase of movement due to the mismatch of compensation control, so the PD regulator plays the main role. With the adaptation of parameter estimates, the quality of compensation control improves, which results in the lowering of tracking errors, thus reducing the signals generated by the PD regulator. It should be noted that the trajectory of the active part of the system is generated based on parameter estimates and, therefore, it is mismatched at the beginning of the movement; then, it is adjusted with the progressive adaptation. This influences the tracking errors of the passive part of the system. The calculation of the control law being used does not require knowledge of the parameters of the mathematical model (inertia, attenuation, elasticity) but only of the model structure. Knowledge of the state of the system (displacements and velocities) and interference is also necessary.

## 7. Simulation Research

Simulation tests consisted of simulating the behavior of the rehabilitation device based on the equations of movement and control laws discussed in previous sections. The data used for simulations are given in Table 1.

The trajectory of the passive part of the system results from the medical support in reference to walking rehabilitation. Recommendations are that rehabilitated patients should be supported in order to enable them to focus on walking correctly. Therefore, the BWS system should maintain a constant rope tension, in spite of the vertical movement made by the patient’s torso when walking. This is what determines the form of the trajectory of the passive part of the system.
(100)$$\{\begin{array}{c} \theta_{pd}=const.=300\;N\\ {\dot{\theta }}_{pd}=0\;\frac{N}{s}\\ {\ddot{\theta }}_{pd}=0\;\frac{N}{s^{2}} \end{array}$$

For the tests, the interference resulting from the vertical movement of the patient’s torso when walking was used. This was determined during experimental research in a rehabilitation center for patients who had suffered a stroke. The average trajectory of movement of their center of mass is described by the following equations:(101)$$\{\begin{array}{c} z_{p}=0.008\left(\sin\left(2\pi t\right)-\sin\left(4\pi t\right)\right)\;m\\ {\dot{z}}_{p}=0.016\pi \left(\cos\left(2\pi t\right)-\cos\left(4\pi t\right)\right)\;\frac{m}{s}\\ {\ddot{z}}_{p}=-0.032\pi^{2}\left(\sin\left(2\pi t\right)-\sin\left(4\pi t\right)\right)\;\frac{m}{s^{2}} \end{array}$$

The interference is presented in Figure 5. It results in the fact that maintaining a constant rope tension requires controlling the movement of the active part of the system according to the trajectory that is generated on an ongoing basis during the device movement.

The simulations considered both the presence and the absence of compensation for interference. Lack of such compensation means that control quality deteriorates. Compensation for the interference requires knowledge of the displacement, velocity and acceleration of the movement of the torso. Under real-life conditions, this requires using at least two sensors, a displacement sensor to determine the position of the torso and an accelerometer to measure its acceleration. The velocity may then be calculated by differentiation of the displacement in time or integration of acceleration. The problems that occur during such processing of the signal are commonly known. Differentiation leads to an increased share of noise in the signal, while integration causes signal drift. Therefore, both double differentiation and integration should be avoided. With reference to the above, the simulation tests included testing another approach, which took into account partial compensation of the interference that only requires knowledge of torso displacement. Such an approach is justified by the fact that the velocity and acceleration of the torso are small and do not generate significant forces that would interfere with the tension force of the rope. Displacement, however, is of key importance. The simulation conditions are summarized in Table 2.

The experiment tested the application of the control algorithm, based on the mathematical model, and of adaptive control. The equations describing the movement of the system, as well as control and adaptation laws used in the simulations, are presented in Appendix A. The main results are presented and discussed in this section, in order to compare methods.

Figure 6 shows the line reaction force (trajectory of the passive part of the system) and the reaction force obtained for different variations of interference compensation using the analyzed control algorithms. Figure 6a shows the reaction force calculated by the control system based on the mathematical model, while Figure 6b shows the reaction force calculated by the adaptive model.

Each of the algorithms gives similar results for total and partial compensation of interference; the courses for variants 1 and 2 are almost identical. In these cases, the rope reaction force approaches the set value. In the variant without interference compensation, better results are obtained using the adaptive algorithm, since it tends to reduce control errors. Nevertheless, lack of knowledge of the interference has a decidedly negative influence on the rope reaction force, causing pulsation of the rope. The results presented confirm the thesis that considering torso displacement is of key importance from the point of view of interference compensation.

The total control signals are shown in Figure 7. For the first two variants of interference compensation, there are no significant differences between the overall control signals. In the variant without interference compensation, the course of control generated by the system based on the mathematical model is smoother than that for the adaptive model. In the latter case, control is characterized by rapid changes, particularly in the initial phase of operation. This is reflected in the course of tracking errors that are presented in Figure 8 and Figure 9.

The tracking errors do not differ significantly for total and partial interference compensation and for both control algorithms. However, for lack of compensation of interference in the adaptive model, tracking errors are characterized by relatively large fluctuations in the initial phase of system operation due to the mismatch of the compensation control based on the adapted parameters. Later, as the parameters are adapted, the fluctuations in tracking errors start to diminish. Unfortunately, this does not take place in the system based on the mathematical model.

Figure 10 presents the estimated parameters of the mathematical model without the compensation of disturbance. Values that deviate significantly from those used in the model of the system resulting from theoretical estimation were used as the initial values of estimated parameters. This simulated the operation of the control algorithm in an actual case when there is an uncertainty of the parameters. According to the proof, parameter estimates remain limited.

## 8. Experimental Results

The experiments involved tests of the rehabilitation device with the use of a control system based on the mathematical model and adaptive control system. Data used in the experimental research are presented in Table 3.

The trajectory of the passive part of the system foresees a phase of tightening the rope, supporting the weight of the patient with a constant force while walking, and a phase of releasing the rope. Smooth changes in the rope tension force should ensure a comfortable interaction between the device and the patient. The given supporting load is described by the function
(102)$$\theta_{pd}=F_{0}+\frac{F_{d}}{1+exp\left[-c\left(t-t_{1}\right)\right]}-\frac{F_{d}}{1+exp\left[-c\left(t-t_{2}\right)\right]}\;N$$
where $F_{0}$ is the constant component of the force with low value that ensures constant tension of the rope and harness supporting the patient, $F_{d}$ is the value of the variable component of the force, c is the coefficient that is responsible for the pace of changes in the force in the rope tightening and releasing phases and $t_{1}$ and $t_{2}$ are the times of rope tightening and releasing, respectively.

The experimental tests took into account the compensation of disturbances in the variant that involves the measurement of the patient’s torso (Figure 11). Figure 12 shows the set rope reaction force (trajectory of the passive part of the system) and the tension force obtained for two variants of the analyzed control algorithms. For the adaptive algorithm, the line tension force is more consistent with the set force. The maximum error from the set value of the force while walking did not exceed 15 N. This result is better in comparison with commercially used devices and comparable with the results obtained in newly published research works. For an algorithm based on a mathematical model, this error was about 25 N. Accurate maintenance of the set force value is impossible in this type of devices, due to the vertical displacements of the torso during walking. The oscillations around the set value visible in the graphs are caused by exactly the movements of the torso, which we interpret as disturbances. Figure 13 shows the trajectory of the active part of the system, i.e., the rotation angle of the motor shaft. The set trajectories differ slightly for each algorithm, because they are generated on an ongoing basis, depending on the trajectory of the passive part.

The total control signals are presented in Figure 14. The highest amplitudes of signals were noted in the transition phases, i.e., during tightening and releasing of the rope. This results from the highest values of tracking errors in these periods (Figure 15). Tracking errors are noticeable lower when the adaptive algorithm is used, in particular a few seconds after starting the system, because the control system has adapted the parameters. The estimated parameters of the mathematical model in the adaptive system are presented in Figure 16. Values that resulted from theoretical calculations were used as the initial values of estimated parameters.

According to the proof of stability, parameter estimates remain limited. Tracking errors are also limited, but not asymptotically convergent to zero, because the real-life system will always be burdened with interferences. As a result, tracking errors may, at the best, be near to zero.

The designed control system strives to maintain the set force regardless of the torso movement of the rehabilitated patient. This has its advantages, as it ensures body weight support with a set force. However, it may also have some disadvantages. For example, if the rehabilitated patient is unable to stand on their own and start to sit down, the system will be unable to prevent it, as it cannot compensate for the total body weight of a human. This is illustrated in Figure 17.

As it can be seen in Figure 17b, the patient’s torso lowers significantly in the 10–15th seconds, while the rope tension force (Figure 17a) is maintained on the set level.

## 9. Conclusions

The authors of the study developed and tested two algorithms to control a BWS system. The algorithms are stable, which is particularly important if the device is applied for human rehabilitation. Their stability was proven theoretically and illustrated by simulation and experimental test results.

Two control algorithms are proposed in the paper. Both are based on a mathematical model; one requires knowledge of the model structure and parameters and the other (adaptive) requires only knowledge of the model structure. In the case under consideration, factors such as the computational complexity of the algorithm, or the measurement availability of the variables describing the system were not a challenge. A stationary rehabilitation device can be equipped with any number of sensors and a system ensuring real-time calculations. For this reason, when selecting the control algorithm, it was taken into account that there are many mathematical formalisms to describe the dynamics of mechanical systems, ensuring the correct structure of the model.

As a result of applying a passive shock absorbing system, the controlled device belongs to the UMS class. Controlling such systems is a complex issue, even more so as the aim is to control the force in the passive part of the system, not the position, as usually happens in UMS systems. Additionally, one of the components of the device is a rope, which only carries a tensile load, so it has to remain taut all the time. If the rope is not taut, the simulation results would not reflect the actual functionality of the system. The results presented in Figure 6 demonstrate that the line reaction force is higher than zero for the time greater than zero, so the rope remains taut.

The control laws presented enable the realization of the force in spite of the fact that one of the elements is passive, because the trajectory of the active element takes into account the dynamic properties of the passive element. The calculation of the control laws used requires knowledge of the model structure and state of the system. The research conducted demonstrates that knowledge of the torso displacement enables very good compensation of interference caused by human motion. In this case, the system trajectory is comparable to the trajectory for full compensation of interference. This result enables a simple system to be used to measure the vertical displacement of the torso in a real-life situation.

Verification tests were also performed on an actual rehabilitation station, including selecting control system parameters and testing algorithms for various interferences caused by human motion. The conducted tests demonstrate that the device realizes the control purpose, which is to maintain the set rope tension, thus, the set weight support force, very well. However, due to certain safety reasons, it is required that the device should be able to increase the weight support force in cases when the human torso is excessively lowered. This will protect the rehabilitated patient from falling down. Due to that, further research will be conducted in order to develop a control algorithm that will realize the set force supporting body weight and support torso displacement according to the adopted movement pattern at the same time.

## Figures and Tables

**Figure 1 sensors-21-05051-f001:**
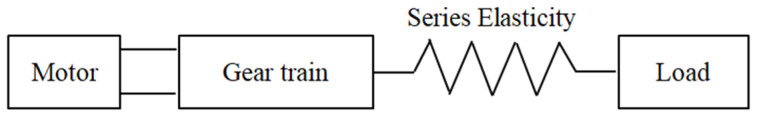
Concept of the series elastic actuator drive.

**Figure 2 sensors-21-05051-f002:**
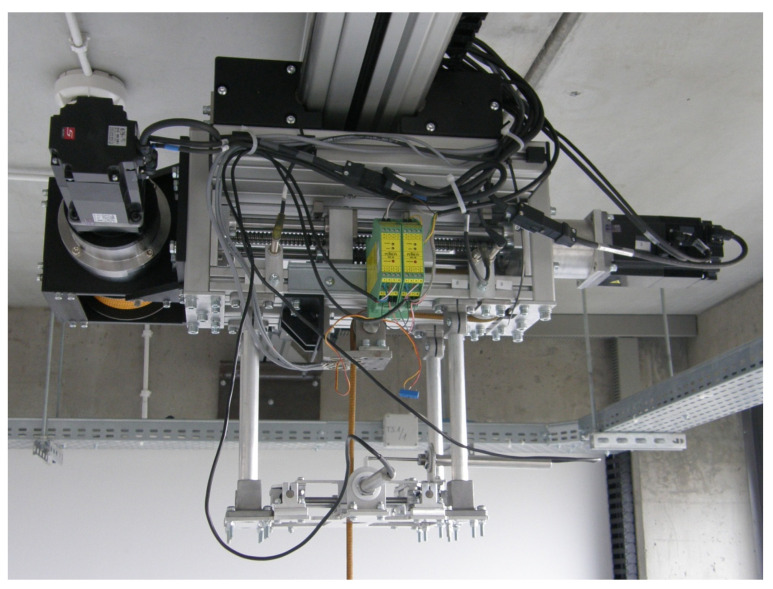
Considered body weight support system.

**Figure 3 sensors-21-05051-f003:**
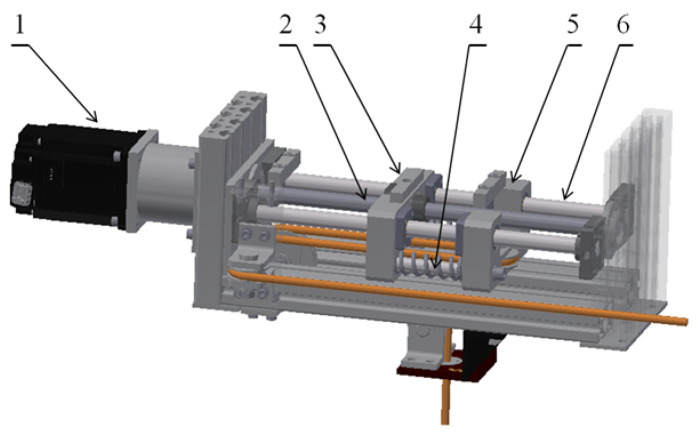
Visualization of the dynamic patient weight compensation system.

**Figure 4 sensors-21-05051-f004:**
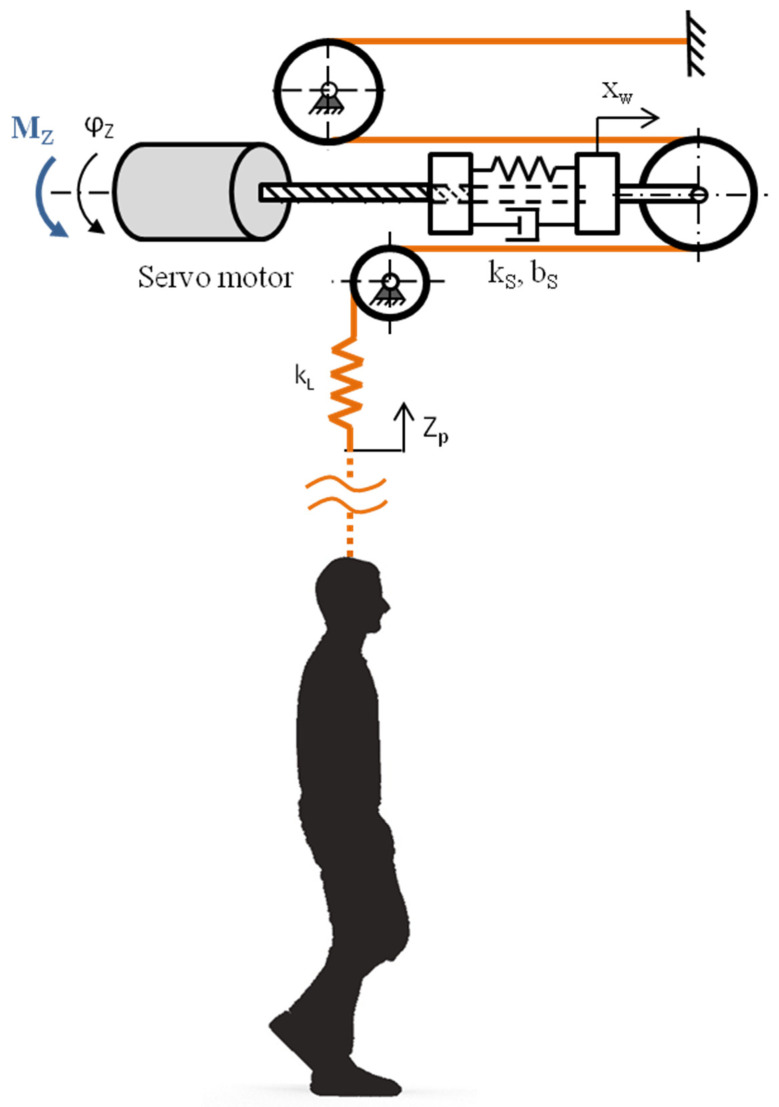
Physical model of the body weight support system.

**Figure 5 sensors-21-05051-f005:**
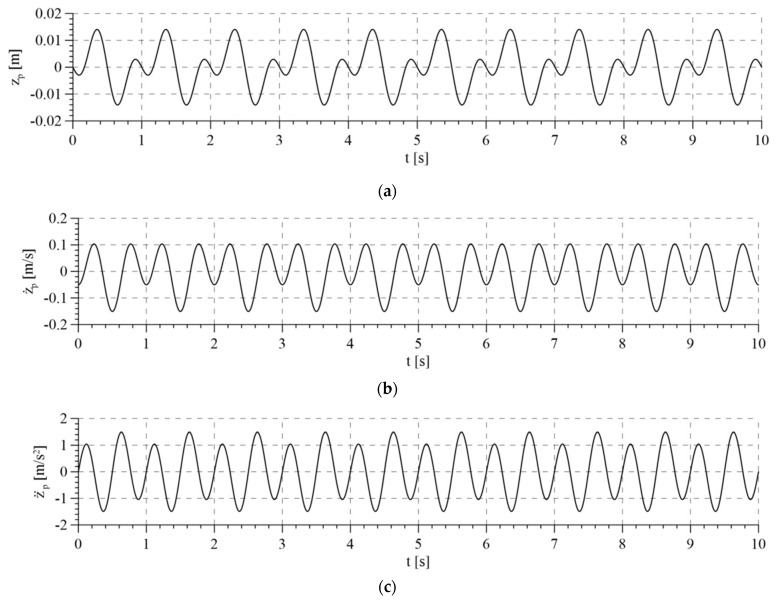
Interferences of the system: (**a**) displacement of torso; (**b**) velocity of torso; (**c**) acceleration of torso.

**Figure 6 sensors-21-05051-f006:**
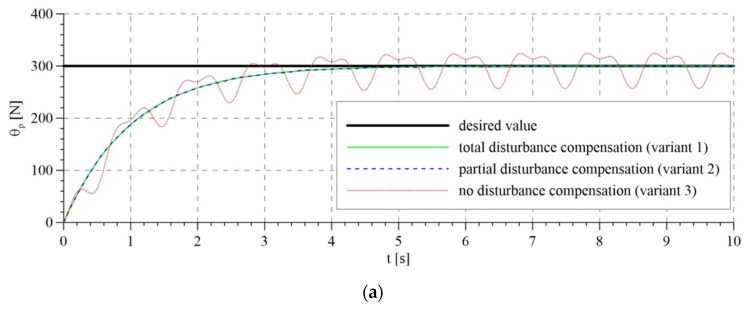
Rope reaction force: (**a**) calculated by the algorithm based on the mathematical model; (**b**) calculated using the adaptive algorithm.

**Figure 7 sensors-21-05051-f007:**
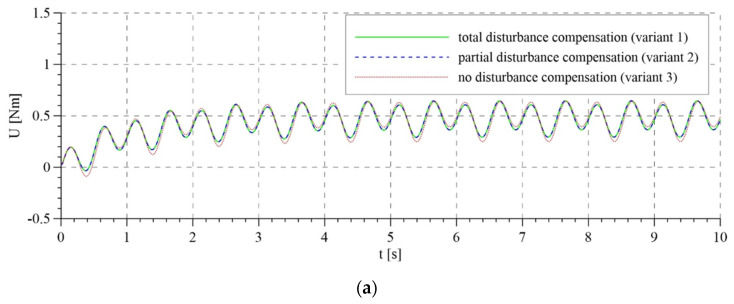
Overall control: (**a**) for the algorithm based on the mathematical model; (**b**) for the adaptive algorithm.

**Figure 8 sensors-21-05051-f008:**
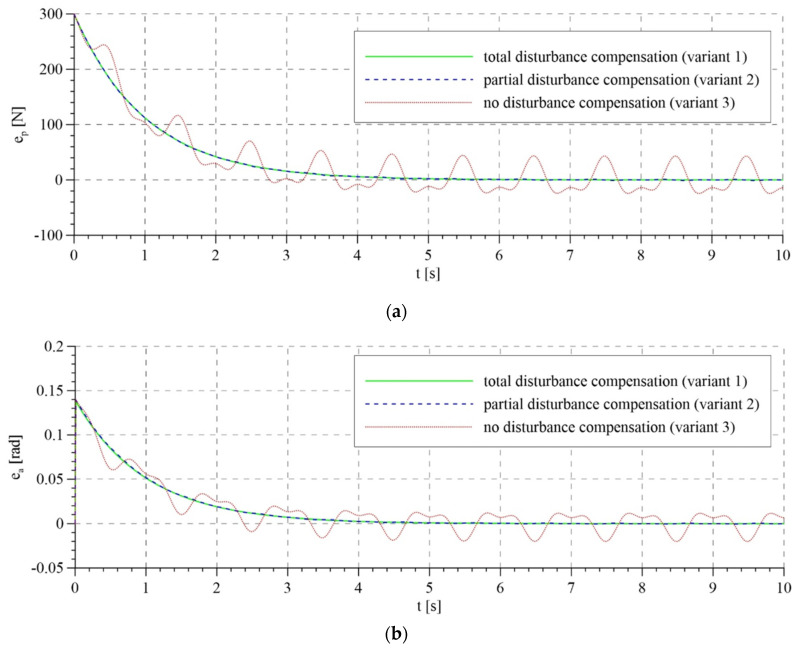
Tracking errors for the control system based on the mathematical model: (**a**) tracking error of the passive part of the system—force error; (**b**) tracking error or the active part of the system—rotation angle error.

**Figure 9 sensors-21-05051-f009:**
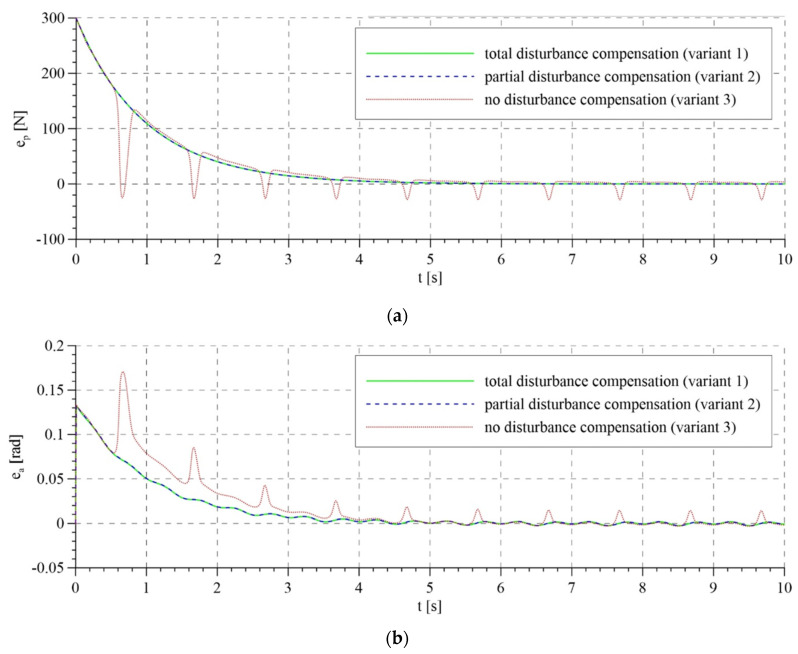
Tracking errors for the adaptive control system: (**a**) tracking error of the passive part of the system—force error; (**b**) tracking error of the active part of the system—rotation angle error.

**Figure 10 sensors-21-05051-f010:**
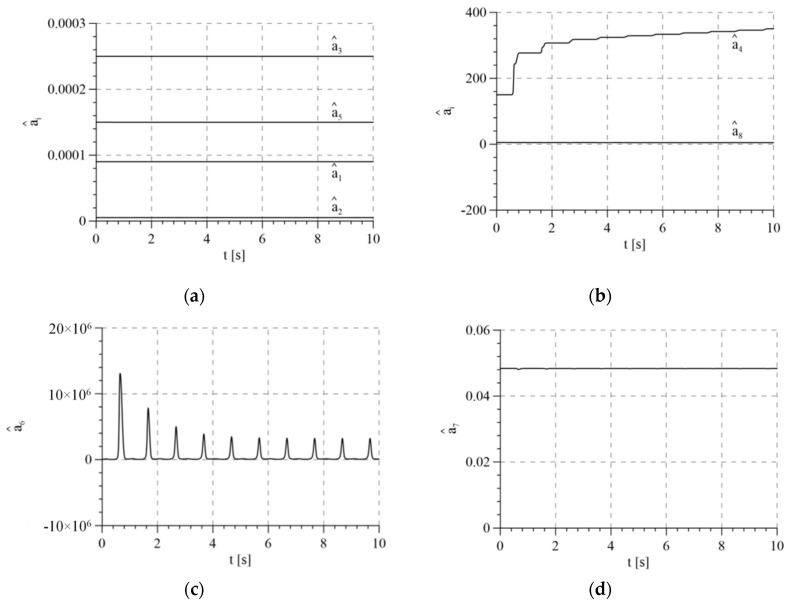
Estimates of model parameters for variant 3: (**a**) estimates of parameters ${\hat{a}}_{1}$, ${\hat{a}}_{2}$, ${\hat{a}}_{3}$ and ${\hat{a}}_{5}$; (**b**) estimates of parameters ${\hat{a}}_{4}$ and ${\hat{a}}_{8}$; (**c**) estimate of parameter ${\hat{a}}_{6}$; (**d**) estimate of parameter ${\hat{a}}_{7}$.

**Figure 11 sensors-21-05051-f011:**
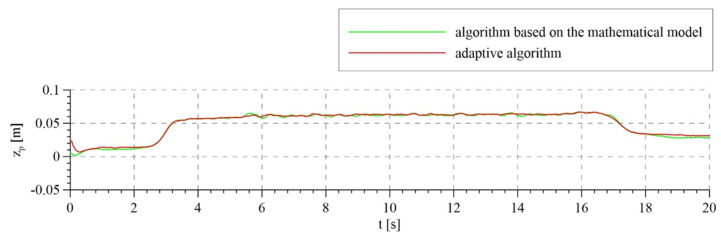
Disturbance of the system—displacement of the patient’s torso.

**Figure 12 sensors-21-05051-f012:**
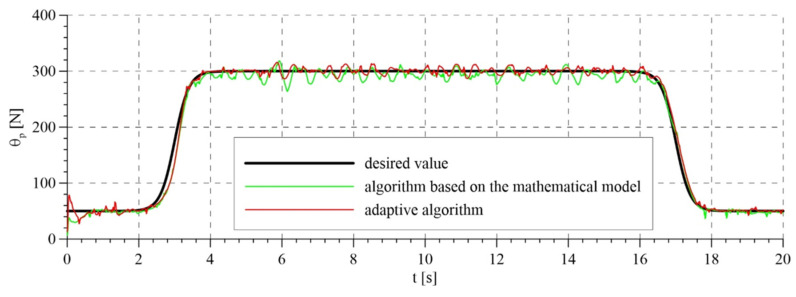
Rope reaction force realized with the use of the algorithm based on the mathematical model and the adaptive algorithm.

**Figure 13 sensors-21-05051-f013:**
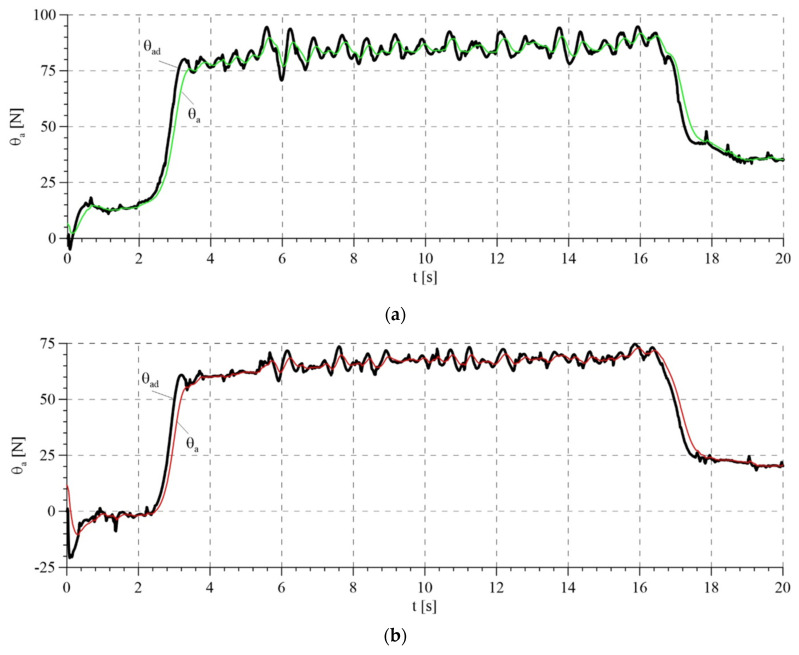
Trajectory of the active part of the system: (**a**) for the algorithm based on the mathematical model; (**b**) for the adaptive algorithm.

**Figure 14 sensors-21-05051-f014:**
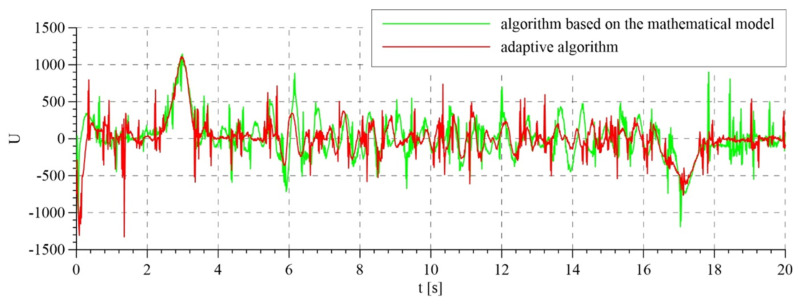
Overall control signals for algorithm based on the mathematical model and for the adaptive algorithm.

**Figure 15 sensors-21-05051-f015:**
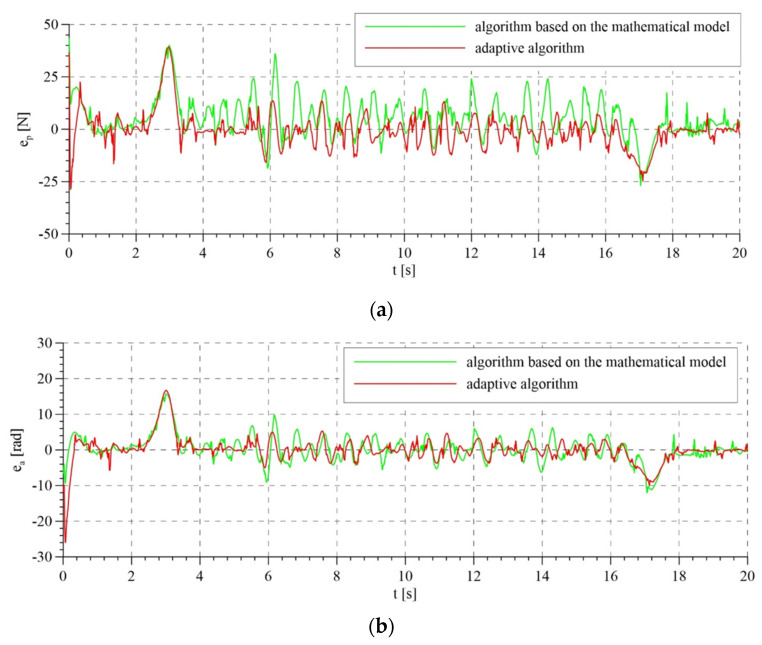
Tracking errors: (**a**) tracking error of the passive part of the system—force error; (**b**) tracking error or the active part of the system—rotation angle error.

**Figure 16 sensors-21-05051-f016:**
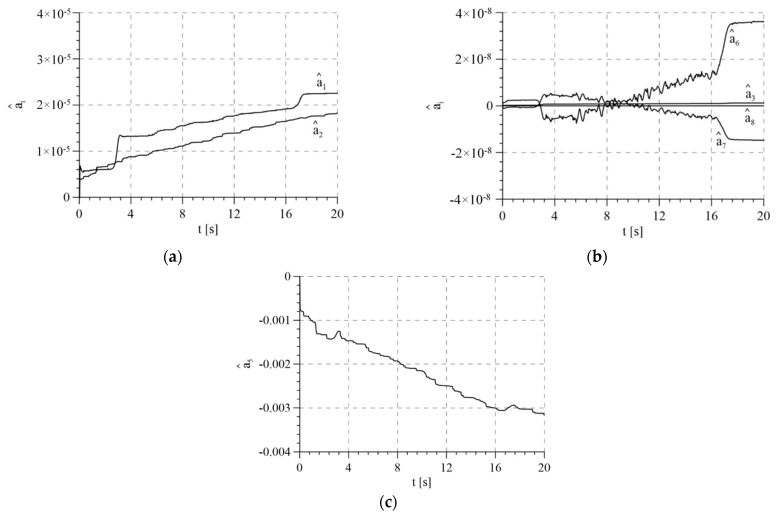
Estimate model parameters in the adaptive system: (**a**) estimates of parameters ${\hat{a}}_{1}$ and ${\hat{a}}_{2}$; (**b**) estimates of parameters ${\hat{a}}_{3}$, ${\hat{a}}_{6}$, ${\hat{a}}_{7}$ and ${\hat{a}}_{8}$; (**c**) estimate of parameter ${\hat{a}}_{5}$.

**Figure 17 sensors-21-05051-f017:**
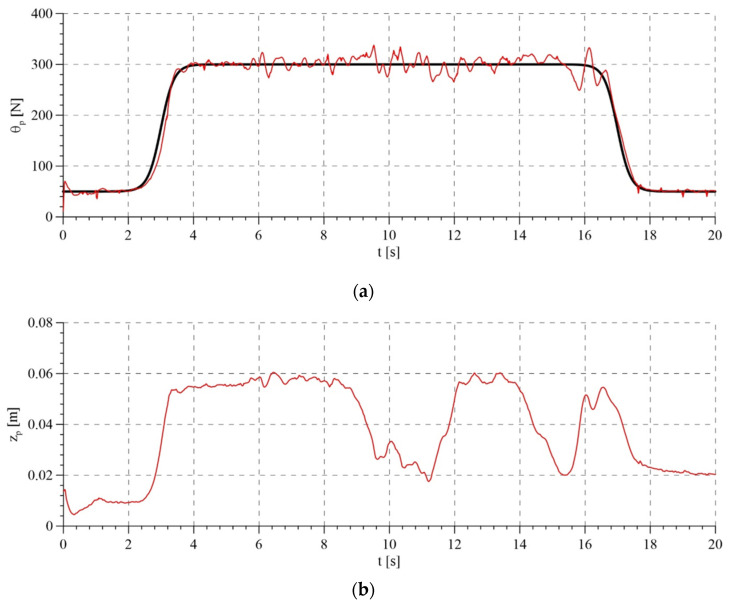
Results of experimental research: (**a**) set and realized rope reaction force with the use of the adaptive algorithm; (**b**) displacement of human torso.

**Table 1 sensors-21-05051-t001:** Parameters of the model and of control systems used in simulation research.

Model Parameters			Control System Parameters in the Model-Based Algorithm and Adaptive Algorithm			Strengthening Adaptation	
Parameters	Unit	Value	Parameters	Unit	Value	Strengthening Adaptation	Value
$I_{Z}$	kg m^2^	0.00018	$\Lambda_{a}$	1/s	1	$\gamma_{1}$	2.1 × 10^−15^
$m_{Z}$	kg	10.5	$\Lambda_{p}$	1/s	1	$\gamma_{2}$	1.8 × 10^−18^
$b_{Z}$	kg m^2^/s	0.0005	$K_{Da}$	kg m^2^/s	3.6	$\gamma_{3}$	2.8 × 10^−14^
$b_{s}$	kg/s	300	$K_{Dp}$	1/s	2.1	$\gamma_{4}$	7.8
$k_{s}$	N/m	96,800				$\gamma_{5}$	7.8 × 10^−15^
$k_{l}$	N/m	1,000,000				$\gamma_{6}$	1.5 × 10^7^
h	m	0.005				$\gamma_{7}$	1.5 × 10^−8^
						$\gamma_{8}$	0.0018

**Table 2 sensors-21-05051-t002:** Variants of the simulated operating conditions of the rehabilitation device.

Variant	Description of Conditions
Variant 1	Total compensation of interference While walking, the rehabilitant uses a model movement given by Equation (101) and the kinematic parameters of the movement of the torso are known.
Variant 2	Partial compensation of interference While walking, the rehabilitant uses a model movement given by Equation (101) and the kinematic parameters of the movement of the torso are known.
Variant 3	Lack of compensation of interference While walking, the rehabilitant uses a model movement given by Equation (101) and the kinematic parameters of the movement of the torso are unknown.

**Table 3 sensors-21-05051-t003:** Parameters of the model and of control systems used in experimental research.

Control System Parameters in the Model-Based Algorithm and Adaptive Algorithm			Strengthening Adaptation	
Parameters	Unit	Value	Strengthening Adaptation	Value
$\Lambda_{a}$	1/s	1	$\gamma_{1}$	2.1 × 10^−15^
$\Lambda_{p}$	1/s	1	$\gamma_{2}$	1.8 × 10^−18^
$K_{Da}$	kg m^2^/s	3.6	$\gamma_{3}$	2.8 × 10^−14^
$K_{Dp}$	1/s	2.1	$\gamma_{4}$	7.8
			$\gamma_{5}$	7.8 × 10^−15^
			$\gamma_{6}$	1.5 × 10^7^
			$\gamma_{7}$	1.5 × 10^−8^
			$\gamma_{8}$	0.0018

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
