# Peer review of "Control System Design of an Underactuated Dynamic Body Weight Support System Using Its Stability"

_sensors, 2021, doi:10.3390/s21155051_

Round 1

Reviewer 1 Report

The introduction/background is well presented, reviewing several relevant and similar studies. However the paper needs minor corrections before publication.

At line 137 is written “PD, PI or PD regulators.”  Maybe one PD should be PID.

The first paragraph from chapter 2 is from template and should be removed.  The same for the first paragraph from chapter 4.

Why PD  control was assumed, why was not try PI and PID since they are widely used in various applications with good performances?

Author Response

Response to the Comments of the First Reviewer

The authors would like to thank the first Reviewer for his/her comments that helped to improve the quality of the paper. The authors’ response to the comments of the first Reviewer is summarized below.

  1. At line 137 is written “PD, PI or PD regulators.” Maybe one PD should be PID.

Thank you for scrupulously reading the manuscripts and listing editing errors.

Of course, on line 137 it was supposed to contain the term "PID".

  1. The first paragraph from chapter 2 is from template and should be removed. The same for the first paragraph from chapter 4.

We also removed paragraphs from the template that were overlooked when formatting the manuscript

  1. Why PD control was assumed, why was not try PI and PID since they are widely used in various applications with good performances?

Referring to the above comment, we want to justify why we adopted the control system with PD controller.

Integral gains used in PI or PID controllers give good results mainly in systems where changes are slow and it is important to reduce the steady state error. Thus, in steady state with constant extortion, a zero error can be obtained, but the time to reach zero is long. Moreover, in the presented case, the dynamics of signal changes is significant because the system is stimulated by interference from a walking person. Initial tests showed that integrating gain does not significantly improve the performance quality, but significantly changes the approach to stability analysis. Mainly for these reasons, the PD regulator was used.

The authors would like to thank again the first reviewer for his/her constructive criticism that led to an improvement the quality of the revised manuscript.

Reviewer 2 Report

The mathematical presentation of this work looks correct, although there is no novelty in the control theories. The reviewer has some concerns regarding the simulation result section.

Why there are so many oscillations in Figure 12, 14 and 17? Is it practical for the control input with large oscillations?

Author Response

Response to the Comments of the Reviewer 2

The authors would like to thank the Reviewer 2 for his/her comment that helped to improve the quality of the paper. The authors’ response to the comment of the Reviewer 2 is below.

  1. Why there are so many oscillations in Figure 12, 14 and 17? Is it practical for the control input with large oscillations?

Accurate control of the set value of the support force is very difficult in this type of devices due to the vertical displacement of the torso during walking. The body weight support system is constantly forced by the training person. So the oscillations around the set value visible in the graphs are caused by the movements of the torso, which we interpret as disturbances. The course of the approximate trajectory of the patient's center of gravity movement, which was used in the simulation tests, is shown in Figure 5.

A commentary on these oscillations is added on page 23, before Figure 11.

The authors would like to thank again the Reviewer 2 for his/her constructive criticism that led to an improvement the quality of the revised manuscript.

Reviewer 3 Report

This paper discusses the stability of systems controlling patient body weight support systems which are used in gait re-education. These devices belong to the class of underactuated mechanical systems. This is due to the application of elastic shock-absorbing connections between the active part of the system and the passive part which impacts the patient. 

Method
The model takes into account properties of the system such as inertia, attenuation and susceptibility to the elements. Stability is an essential property of the system due to human-device interaction. In order to demonstrate stability, Lyapunov’s theory of stability was applied which is based on the model of  system dynamics. The stability of the control system based on a model that requires knowledge of the structure and parameters of the equations of motion was demonstrated. Due to inaccuracies in the modeling of the rope (one of the basic elements of the device) an adaptive control system was introduced and its stability also was proved.

Results:
The authors conducted simulation and experimental tests that illustrate the functionality of the analyzed control systems.

This work is a solid work from the modelling part till the control part. It even contains experiments. I think the work is of high quality and well organized. The results are also extremely interesting, i.e. they show the benefits of adaptation. I only have minor comments

The authors are missing some recent works on underactuated mechanical systems (with adaptive control)

Towards structure-independent stabilization for uncertain underactuated Euler–Lagrange systems
Automatica23 December 2019...
Spandan RoySimone Baldi

Adaptive tracking control for underactuated mechanical systems with relative degree two
Automatica13 April 2021...
Manuel GnucciRiccardo Marino

Some equations are too long and should be split in two lines

Stability of a closed system -> Stability of a closed-loop system?

Author Response

Response to the Comments of the Reviewer 3

The authors would like to thank the Reviewer 3 for his/her comment that helped to improve the quality of the paper. The authors’ response to the comments of the Reviewer 3 is below.

  1. The authors are missing some recent works on underactuated mechanical systems (with adaptive control)

We agree that there was little information in the manuscript about the use of adaptive systems. Therefore, in the introduction we added information about the effectiveness of using adaptive systems in controlling the underactuated mechanical systems, including references to the proposed articles. The justification why the adaptation system was needed in our work can be found in the last paragraph of the introduction.

      2. Some equations are too long and should be split in two lines

All equations in manuscript were saved using the editor template. The template contains two ways of writing equations, in a shorter version (with a width equal to a paragraph) and in a longer version (and this is what we used in some places). However, where it was possible, the equations were split and moved to the next line.

      3. Stability of a closed system -> Stability of a closed-loop system?

Of course, unfortunately we used the expression “closed system”. This mistake has been corrected in all places.

The authors would like to thank again the Reviewer 3 for his/her constructive criticism that led to an improvement the quality of the revised manuscript.